# Long-Term Pregnancy Outcomes of Patients with Diffuse Adenomyosis after Double-Flap Adenomyomectomy

**DOI:** 10.3390/jcm11123489

**Published:** 2022-06-17

**Authors:** Yong Zhou, Li Shen, Yuan Wang, Mengjia Yang, Zhengyun Chen, Xinmei Zhang

**Affiliations:** Women’s Hospital, Zhejiang University School of Medicine, No.1 Xueshi Road, Hangzhou 310006, China; zhouyongforce@zju.edu.cn (Y.Z.); 5517041@zju.edu.cn (L.S.); 5512042@zju.edu.cn (Y.W.); 5516024@zju.edu.cn (M.Y.); chenzy1290@zju.edu.cn (Z.C.)

**Keywords:** diffuse adenomyosis, double-flap adenomyomectomy, fertility-sparing surgery, cumulative pregnancy rate, junctional zone

## Abstract

Although many studies show that patients with diffuse adenomyosis who underwent fertility-sparing surgery can have a successful pregnancy, their pregnancy outcomes are still controversial. The objective of this study was to determine long-term pregnancy outcomes and possible influencing factors after double-flap adenomyomectomy for patients with diffuse adenomyosis. A total of 137 patients with diffuse adenomyosis who underwent double-flap adenomyomectomy between January 2011 and December 2019 were studied, and correlations between pregnancy outcomes and clinical data, including age and junctional zone measured by magnetic resonance imaging (JZ_max-A_), were analyzed. The results show that 56 patients (40.9%, 56/137) had 62 pregnancies, including 35 natural pregnancies and 27 assisted reproduction pregnancies, after operation. A univariate regression analysis showed that the pregnancy outcomes were related to age at surgery, visual analog scale (VAS) score of preoperative dysmenorrhea, parity experience, length of infertility, and postoperative JZ_max-A_. A multivariate regression analysis showed that age at surgery, VAS score of preoperative dysmenorrhea, and postoperative JZ_max-A_ were the independent indicators correlated with pregnancy outcomes. A receiver operating characteristic curve analysis showed that postoperative JZ_max-A_ was the most valuable indicator for predicting pregnancy outcomes. Cumulative pregnancy rates during the first 3 years were 70.1% and 20.9% in the postoperative JZ_max-A_ ≤ 8.5 mm and the postoperative JZ_max-A_ > 8.5 mm groups, respectively. In conclusion, double-flap adenomyomectomy could improve fertility for diffuse adenomyosis, and postoperative JZ_max-A_ might be a promising indicator for predicting pregnancy outcomes.

## 1. Introduction

Adenomyosis is a frequent benign gynecological disorder characterized by heterotopic endometrial glands and stroma that invade the myometrium. Adenomyosis in infertile women has increased in recent years owing to the prevalence of high-resolution ultrasound and magnetic resonance imaging (MRI), as well as the increasing age of women seeking fertility treatment [1,2]. Although the impact of adenomyosis on pregnancy outcomes is controversial, the accessible evidence increasingly [3,4] shows that adenomyosis has a detrimental effect on fertility. At present, the beneficial effect of the surgical treatment of adenomyosis on fertility is still disputable. Dueholm et al. [5] included a sub-analysis of reproductive outcomes after fertility-sparing surgery among 338 women with adenomyosis, and they noted markedly higher pregnancy and live birth rates in patients who underwent surgery. However, two articles, which compared the spontaneous pregnancy rate in the 3 years after treatment with gonadotropin releasing hormone agonist (GnRH-a) alone and with fertility-sparing surgery, found that surgery is associated with an obviously increased pregnancy rate [6,7]. A high-intensity focused ultrasound (HIFU) as a noninvasive treatment could improve the pregnancy outcomes of patients with adenomyosis. Among 93 patients with adenomyosis and infertility [8], patients treated with the HIFU showed a considerably higher pregnancy rate (52%) than those who underwent laparoscopic excision surgery (30.2%). However, diffused lesions in patients with diffuse adenomyosis require larger ablation areas, leading to more severe burning of the uterine endometrium and a reduced non-perfused volume. For these reasons, they might have lower pregnancy rates after the HIFU treatment [9]. Besides, the radiofrequency ablation of adenomyotic lesions is another promising fertility-preserving option for adenomyosis [10]. However, it has the potential risk of collateral thermal damage to the nearby intestinal tract. Therefore, the recommendation of this procedure for patients desiring fertility should be approached with caution.

Focal adenomyosis is often well circumscribed and confined to a limited part of the uterus, and the complete excision of affected lesions is typically easier [11]. Diffuse adenomyosis fares worse than focal adenomyosis on fertility owing to the presence of more extensive lesions, which may compromise uterine function and integrity during pregnancy [8]. However, there are limited studies on pregnancy after the surgical treatment of diffuse adenomyosis. In the few available studies [6,12], among the variety of treatment options for improving the reproductive outcomes of women with diffuse adenomyosis, fertility-sparing surgery has shown varying success (range: 9.4–100%). Several studies reported that age, type of adenomyosis, postoperative GnRHa application, and surgical methods may affect fertility outcomes following uterus-sparing surgery for uterine adenomyosis [13,14]. However, there is still a lack of consensus on the determining factors of successful pregnancy after conservative surgery for diffuse adenomyosis.

Double-flap adenomyomectomy [15,16] has been used to treat diffuse adenomyosis for over 10 years. Among these cases, a sizable group desired pregnancy after surgery. This study aimed to evaluate the reproductive outcomes of patients with diffuse adenomyosis after a double-flap adenomyomectomy and incorporate possible influencing factors into the analysis of fertility outcomes.

## 2. Materials and Methods

### 2.1. Patients

In this retrospective cross-sectional study, we recruited patients diagnosed with adenomyosis who underwent uterus-sparing surgery through a double-flap adenomyomectomy from January 2011 to December 2019. Data, including age, body mass index (BMI), visual analogue scale (VAS) score of dysmenorrhea, symptoms, pregnancy history, and other hospital records, were thoroughly obtained. We followed up with all the patients to collect fertility information either face-to-face or via telephone interviews following surgery. The junctional zone (JZ) was measured by MRI before and after surgery. The JZ is predominantly only visible in the isthmic part of diffuse adenomyosis, and the other parts are replaced by adenomyosis lesions. For this reason, we introduced the term “JZ_max-A_” [17] instead of using JZ, as shown in Figure 1.

### 2.2. Management Strategy after Adenomyomectomy

The management strategy following fertility-sparing surgery was as follows. First, 3–6 courses of gonadotropin releasing hormone agonist (GnRHa) therapy were given to inhibit the growth of residual lesions after surgery. Then, after the GnRHa treatment was over and the first menstruation was restored, an MRI was performed to determine the integrity of the muscle layer, identify whether a hematoma formed at the surgical site, and measure the JZ_max-A_. As the endometrium was opened in double-flap adenomyomectomy secondary to intrauterine adhesions, we performed hysteroscopies to treat possible intrauterine adhesion and observe the intrauterine fallopian tube opening. The surgical procedure required a full-thickness cut of the uterus. A scarred uterus may rupture during pregnancy. Therefore, we recommended that patients prepare for pregnancy at least 1 year after surgery. A cesarean section was recommended as the delivery mode. For the women who became pregnant, clinical pregnancy was only taken into account when the gestational sac was confirmed by an ultrasound examination as part of the study.

### 2.3. Statistical Analysis

A data analysis was undertaken using Graph Pad Prism version 6.00 (Windows, GraphPad software, San Diego, CA, USA). The statistical analysis was based on a Student’s *t*-test or ANOVA. ANCOVA was utilized to adjust to the baseline assessment, and a multivariate logistic regression was provided to assess potential confounding factors. A receiver operating characteristic (ROC) curve was plotted to determine the value of pregnancy prediction, and a Kaplan–Meier (K–M) analysis was used to assess the cumulative pregnancy rate (CPR). *p* < 0.05 was considered significant in all analyses.

## 3. Results

### 3.1. Comparison of Baseline Characteristics between Pregnant and Non-Pregnant Patients

A flow diagram showing the number of women with diffuse adenomyosis, an indication of surgery, and fertility outcomes is set out in Figure 2.

Among the 196 cases who underwent fertility-sparing surgery for diffuse adenomyosis, 59 (30.1%) cases were lost to follow-up or gave up pregnancy after surgery and thus were excluded from the analysis. Among the 137 (69.9%) patients who were finally included, 56 (40.9%) became pregnant, and 81 (59.1%) were non-pregnant. A comparison of the baseline characteristics of the two groups is detailed in Table 1. Remarkable differences in age at surgery, VAS score of preoperative dysmenorrhea, length of infertility, parity experience, and postoperative JZ_max-A_ were found between the pregnant and non-pregnant women. The two groups were similar in BMI, previous medical history (uterine surgery, pelvic surgery, miscarriage, infertility, gravidity experience, and preoperative medication), coexistence with myoma or endometriosis, preoperative JZ_max-A,_ and length of the corpus uteri. The incidence of adhesions was 2.3% (3/137), and both fallopian tube openings in each patient were observed by hysteroscopy.

### 3.2. Reproductive Outcomes after Fertility-Sparing Surgery

The indications for surgery were infertility and severe dysmenorrhea. Among the patients, 49 (35.8%) attempted spontaneous pregnancy, 40 patients underwent assisted reproductive technology (ART) after failed spontaneous pregnancy, and 48 chose ART directly. A total of 62 pregnancies were successfully registered by 56 women, among which 35 (56.5%) were spontaneous pregnancies, six (9.7%) patients underwent ART after a failed spontaneous pregnancy, and 21 (33.9%) underwent ART directly. Among the 114 patients diagnosed with infertility before surgery, 47 (41.2%) cases successfully became pregnant after surgery. The pregnancy outcomes were miscarriage (14, 22.6%), ongoing pregnancy during follow-up (three, 4.8%), and live births (45, 72.6%). Among the 45 live births, 39 (86.7%) were full-term births, and the remaining six (13.3%) were immature births, including two pairs of twins. Moreover, 61 births were delivered by cesarean section, and only one was delivered by vaginal approach. Among the 137 cases in this study, the recurrence rate within 3 years was zero, and only seven (5.1%) of the cases had a recurrence. None of the seven had a successful pregnancy.

Remarkable differences in preoperative ART failure, preoperative parity experience, and miscarriage after surgery were found between women aged <35 years and those aged ≥35 years at surgery (Table 2). The two groups were similar in BMI, preoperative pregnancy outcomes (miscarriage, infertility, and gravidity), and postoperative pregnancy outcomes (natural conception, ART conception, and term live birth). No considerable differences in pregnancy outcomes were found between natural conception and ART conception after surgery.

### 3.3. Possible Indicators for Predicting Pregnancy after Surgery

A multivariate analysis was performed to exclude the influence of the confounding factors set out in Table 3.

The multivariate regression analysis showed that age at surgery, VAS score of preoperative dysmenorrhea, and postoperative JZ_max-A_ were the independent factors that affected postoperative pregnancy outcomes. According to the ROC analysis of the three indicators (Figure 3), postoperative JZ_max-A_ was associated with the highest area under the curve (AUC); therefore, it might be the most valuable indicator for predicting pregnancy outcomes after surgery. A further analysis showed that the best cut-off point of the postoperative JZ_max-A_ score was 8.5 mm (sensitivity: 76.4%, specificity: 75.3%), and the Youden index was 0.517 (Youden index = sensitivity + specificity − 1). We classified these cases into two groups according to the best cut-off point: the postoperative JZ_max-A_ ≤ 8.5 mm group and the postoperative JZ_max-A_ > 8.5 mm group. According to the K−M analysis, CPR was significantly higher in the postoperative JZ_max-A_ ≤ 8.5 mm group than that in the postoperative JZ_max-A_ >8.5 mm group (Log-rank Mantel−Cox test, χ^2^ = 38.14, 95% CI = 3.27–9.86, *p* = 0.001). The CPRs at 24 and 36 months after surgery were 46.6% and 70.1%, respectively, and rose to 79.9% after 60 months. However, the CPRs in the first 2 and 3 years after surgery in the postoperative JZ_max-A_ > 8.5 mm group were 11.5% and 20.9%, respectively, and did not increase in subsequent years.

## 4. Discussion

In this study, we found that fertility performance improved with a higher clinical pregnancy rate and a lower miscarriage rate after adenomyomectomy for patients with diffuse adenomyosis. Age at surgery, VAS score of preoperative dysmenorrhea, and postoperative JZ_max-A_ were strong risk factors for fertility outcomes. Among them, postoperative JZ_max-A_ = 8.5 mm can be used as an important cutoff value to predict CPR.

Counseling in relation to the surgical approach, including the potential benefits and harms of diffuse adenomyosis on reproductive outcomes, should be tailored and individualized [18]. Therefore, we carefully recommended double-flap adenomyomectomy only for patients with diffuse adenomyosis who had a strong desire to preserve fertility. These patients might have suffered from infertility, severe pain, failure of non-surgical treatments, and even ART. The challenge of fertility-sparing surgery is to distinguish the affected lesions from the normal myometrium according to the characteristics of diffuse adenomyosis [19]. As a result, difficulties faced in double-flap adenomyomectomy include the removal of as many lesions as possible and the retention of enough muscle layers to restore the optimal function of the uterine wall. Consequently, the risk of a uterine rupture increases in pregnancy, which seems to be around 6% as reported [3,20]. Fortunately, no patient had suffered a uterine rupture among the 62 pregnancies in this study.

The surgical removal of adenomyosis lesions can reduce the deleterious effects of the disease on fertility. Saremi et al. [21] reported a pregnancy rate of 30% among 70 patients desiring pregnancy after abdominal adenomyomectomy, and 16 (76.2%) patients obtained full-term live births. Another systematic review [22] revealed a mean pregnancy rate of 34.1% in patients with diffuse adenomyosis after fertility-sparing surgery. Among the 137 patients in our study, the pregnancy and full-term live birth rates were 40.9% and 86.7%, respectively, excluding those who were in gestation peroid. Moreover, fertility-sparing surgery has been proven effective for probable adenomyosis-related infertility [5]. Indeed, among the 114 patients with infertility in this study, 47 (41.2%) patients successfully achieved pregnancy. It should be noted that we do not intend to state that our results are superior to other published reports, because our reports were not adjusted to the same protocols owing to differences in data among various studies.

Adenomyosis is closely related to miscarriage [23]. A study reported that the incidence of adenomyosis in 68 patients with miscarriages was 38.2% on ultrasound examination [24]. Consistent with the above studies, our patients experienced a miscarriage rate as high as 56.9% before surgery. However, the postoperative miscarriage rate dropped to 22.6% among these patients. Hence, we believed that an improvement in miscarriage rates was brought about by double-flap adenomyomectomy after surgery. For these reasons, we highlight that management related to double-flap adenomyomectomy could improve the fertility of women with diffuse adenomyosis. Besides, among the six immature births, four newborns comprised two pairs of twins. Therefore, we suggest that singleton pregnancy might be more suitable to reduce the risk of preterm birth after surgery.

The literature on factors influencing successful postoperative pregnancy is scarce. According to the available data, younger patients with diffuse adenomyosis are more likely to benefit from surgery and succeed in pregnancy. Given that the average age at surgery of non-pregnant patients (37.46 ± 3.81 years) was older than that in pregnant patients (33.21 ± 3.72 years) in this study, the most likely reason might be the decrease in the number and quality of oocytes with advancing maternal age [25,26,27]. These findings demonstrate the strong negative influence of age on fertility outcomes, which might imply that double-flap adenomyomectomy cannot compensate for the age-related decline in fertility. Therefore, we propose that young patients who wish to preserve fertility should seize the opportunity in this regard. Moreover, according to the univariate regression analysis, a higher VAS score of preoperative dysmenorrhea, longer time of infertility, and thicker postoperative JZ_max-A_ were observed in non-pregnant women. This finding might suggest that adverse pregnancy events were likely to be higher among patients with diffuse adenomyosis owing to the presence of more extensive lesions, which may compromise uterine function and integrity during pregnancy. Moreover, the preoperative delivery experience may be a potential protective factor for pregnancy after surgery, because more parity was observed in the pregnant group. However, the multivariate regression analysis showed that only age at surgery, VAS score of preoperative dysmenorrhea, and postoperative JZ_max-A_ were the independent indicators that were correlated with pregnancy outcome.

According to the ROC analysis, postoperative JZ_max-A,_ which had the highest AUC, was considered a very promising indicator for predicting pregnancy outcome after surgery. Furthermore, we developed a prediction model of CPR by K–M analysis. For patients with postoperative JZ_max-A_ ≤ 8.5 mm, the CPR reached 70.1% in the first 3 years and rose to 79.9% in the first 5 years. These findings indicated that the optimal time for pregnancy could last 3 years after surgery, but the chance to attain a successful pregnancy was very slim after this period. Nonetheless, for patients with postoperative JZ_max-A_ > 8.5 mm, only 20.9% of the pregnancies occurred in the first 3 years and no patient became pregnant in subsequent years. Thus, these results seemed to be very crucial for clinicians to provide recommendations to predict pregnancy after double-flap adenomyomectomy.

The advantages of this study include the use of univariate and multivariate analysis to identify possible factors affecting postoperative pregnancy. Based on the ROC analysis, we determined postoperative JZ_max-A_ as the most valuable indicator for predicting pregnancy. Finally, a CPR prediction model was constructed using a K–M analysis to provide recommendations for clinical fertility counseling.

However, our study also has several limitations. First, it is a retrospective study, and all the included patients were highly selected. Second, the absence of standardized surgical techniques and differences in surgeons’ skills and experience further contributed to heterogeneity in reproductive outcomes. Lastly, ovarian reserve is also a very important factor for pregnancy. However, this study failed to assess the impact of surgery on ovarian function.

## 5. Conclusions

The principal strength of this study is that double-flap adenomyomectomy could remarkably improve pregnancy for patients with diffuse adenomyosis. Younger age at surgery, lower VAS score of preoperative dysmenorrhea, and thinner postoperative JZ_max-A_ were closely related to postoperative pregnancy. The CPR during the first 3 years was as high as 70.1%when the postoperative JZ_max-A_ was ≤8.5 mm. We believe that these findings may have direct implications for clinical practice when designing individualized fertility treatments, such as the timing of fertility treatments and the possibility of a successful pregnancy. Moreover, the CPR model constructed to predict pregnancy could provide a novel insight into fertility counseling after fertility-sparing surgery.

## Figures and Tables

**Figure 1 jcm-11-03489-f001:**
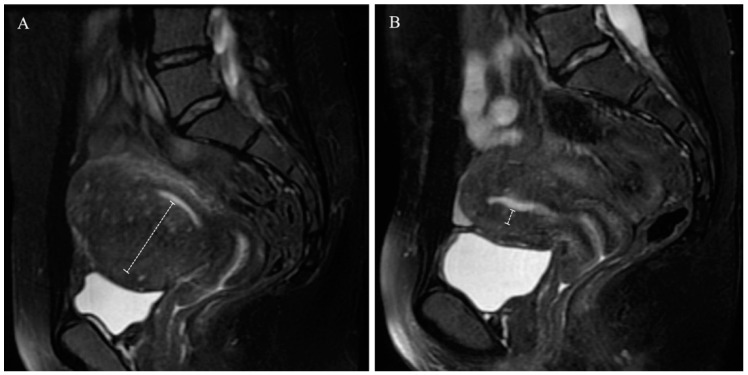
In diffuse adenomyosis, the JZ is irregular or only visible in the isthmus of the uterus, and the other parts are replaced by adenomyotic lesions. Therefore, we introduced the term “JZ_max-A_” instead of using JZ. All low-intensity signal areas representing diffuse circumscribed adenomyosis attached to the JZ are included in JZ_max-A_. (**A**) Preoperative JZ_max-A_, (**B**) postoperative JZ_max-A_, JZ = junctional zone.

**Figure 2 jcm-11-03489-f002:**
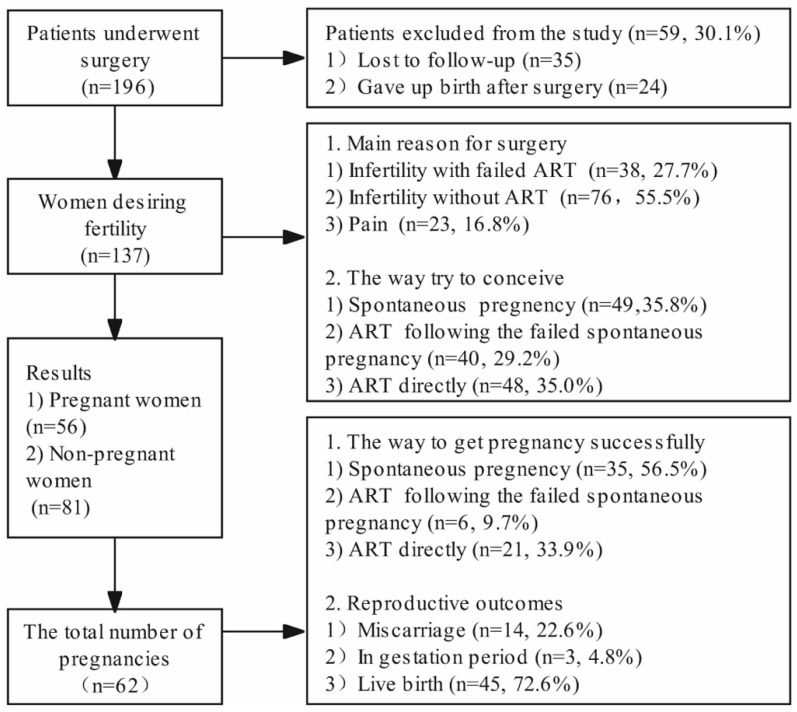
Flow diagram of the clinical data of women who underwent double-flap adenomyomectomy.

**Figure 3 jcm-11-03489-f003:**
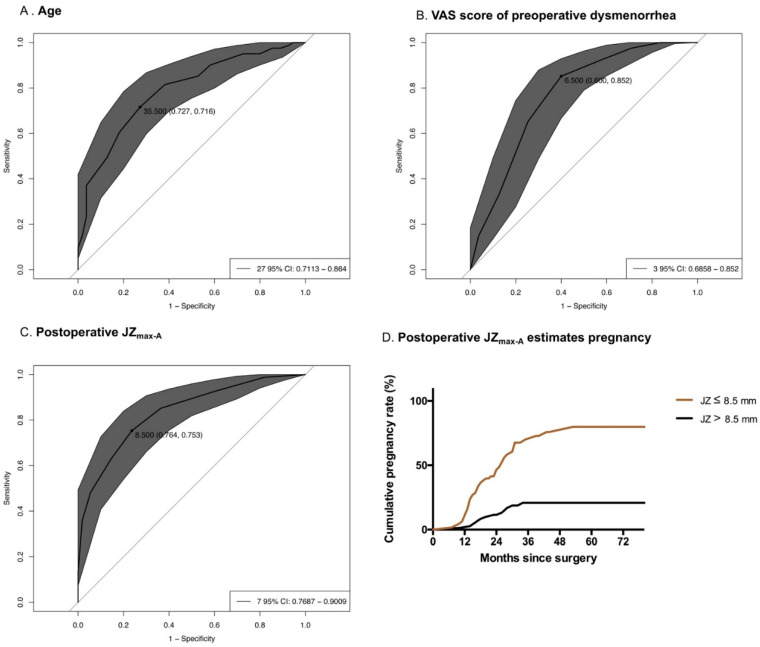
(**A**–**C**) ROC analysis for pregnancy prediction: (**A**) AUC of age at surgery = 0.791, *p* = 0.001; (**B**) AUC of the VAS score of preoperative dysmenorrhea = 0.773, *p* = 0.015; (**C**) AUC of postoperative JZ_max-A_ = 0.836, *p* = 0.001; (**D**) Postoperative JZ_max-A_ estimation of pregnancy. The K–M analysis showed significantly differences between the postoperative JZ_max-A_ ≤ 8.5 mm group and the postoperative JZ_max-A_ > 8.5 mm group (Log-rank Mantel–Cox test, χ^2^ = 38.14, 95% CI = 3.27–9.86, *p* = 0.001).

**Table 1 jcm-11-03489-t001:** Comparison of baseline characteristics between pregnant and non-pregnant patients (*n* = 137).

Variables	Pregnant Patients (*n* = 56)	Non-Pregnant Patients (*n* = 81)	*p*/X^2^ Value	95%CI
**Age at surgery, years**	34 (25, 42)	37 (27, 45)	0.007	2.94~5.54
**BMI, kg/m^2^**	22.75 (18.10, 24.50)	23.10 (18.50, 25.50)	0.658	0.24~0.76
**Preoperative dysmenorrhea (VAS score)**	5.69 ± 2.44	7.89 ± 1.45	0.021	1.53~2.85
**Previous history**				
Uterine surgery	7/56 (12.5%)	16/81 (19.8%)	0.264	0.22~1.52
Pelvic surgery	7/56 (12.5%)	14/81 (17.3%)	0.445	0.26~1.82
Miscarriage	29/56 (76.8%)	49/81 (87.7%)	0.312	0.23~0.99
Infertility	43/56 (76.8%)	71/81 (87.7%)	0.094	0.19~1.15
Length of infertility, years	5.05 ± 1.96	5.92 ± 2.51	0.031	0.15~2.13
Gravidity	43/56 (76.8%)	55/81 (67.9%)	0.257	0.72~3.40
Parity	19/56 (33.9%)	13/81 (16.0%)	0.015	1.19~6.05
Medication	9/56 (30.4%)	15/81 (28.4%)	0.711	0.34~2.09
ART failure	17/56 (30.4%)	23/81 (28.4%)	0.854	0.52~2.32
**Coexist with myoma**	3/56 (5.4%)	11/81 (13.6%)	0.118	0.10~1.37
**Coexist with endometriosis**	24/56 (42.9%)	34/81 (42.0%)	0.918	0.52~2.07
**The JZ_max-A_** **,** **mm**				
Preoperative	44.36 ± 1.06	46.22 ± 0.89	0.911	−0.86~4.62
Postoperative	8.14 ± 1.91	11.23 ± 2.34	0.039	2.26~3.72
Corrected postoperative	7.23 ± 0.29	10.23 ± 0.24	0.001	1.76~6.54
**Length of corpus uteri** **,** **cm**				
Preoperative	9.16 ± 8.91	8.91 ± 0.12	0.086	−0.67~0.16
Postoperative	6.13 ± 0.13	6.39 ± 0.11	0.783	−0.08~0.60
Corrected postoperative	6.04 ± 0.09	6.45 ± 0.07	0.118	0.31~0.65

CI: confidence interval.

**Table 2 jcm-11-03489-t002:** Comparison of pregnancy outcomes between women aged <35 and aged ≥35 years (*n* = 56).

Variables	Aged < 35 Years (*n* = 35)	Aged ≥ 35 Years (*n* = 21)	*p*/X^2^ Value	95%CI
BMI, kg/m^2^	23.00 (19.60, 24.50)	22.10 (18.10, 25.50)	0.184	−1.30~0.25
Preoperative pregnancy outcomes				
ART failure	5/35 (14.3%)	12/21 (57.1%)	0.001	0.04~0.45
Miscarriage	20/35 (57.1%)	9/21 (42.9%)	0.300	0.60~5.31
Infertility	27/35 (77.1%)	16/21 (76.2%)	0.935	0.29~3.78
Gravidity experience	26/35 (74.3%)	17/21 (81.0%)	0.567	0.18~2.56
Parity experience	14/35 (40.0%)	5/21 (23.8%)	0.027	0.64~7.16
Postoperative pregnancy outcomes				
Natural conception	22/38 (57.9%)	13/24 (54.2%)	0.773	0.42~3.26
ART conception	16/38 (42.1%)	11/24 (45.8%)	0.773	0.31~2.41
Miscarriage	5/38 (13.2%)	9/24 (37.5%)	0.026	0.07~0.88
In gestation period	1/38 (2.6%)	2/24 (8.3%)	0.308	0.03~−3.47
Live birth				
Preterm	2/38 (5.3%)	0		
Term	30/38 (78.9%)	13/24 (54.2%)	0.100	1.04~9.72
Gestational age	37.99 ± 0.13	38.11 ± 0.11	0.575	−0.32~0.57

**Table 3 jcm-11-03489-t003:** Multivariate analysis of cofounding factors for postoperative pregnancy.

Variables	β	Exp(β)	95%CI	*p*
Age of surgery	−0.082	0.921	0.809~1.049	0.014
Preoperative dysmenorrhea (VAS score)	−0.672	0.511	0.473~0.697	0.001
Preoperative parity	0.485	1.623	0.515~5.121	0.408
Length of infertility	−0.177	0.838	0.674~1.042	0.112
Postoperative JZ_max-A_	−0.608	0.545	0.424~0.701	0.001

## Data Availability

The data underlying this article cannot be shared publicly to protect privacy of the individuals that participated in the study. Data will be shared upon reasonable request to the corresponding author.

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
