# Peer review of "Long-Term Pregnancy Outcomes of Patients with Diffuse Adenomyosis after Double-Flap Adenomyomectomy"

_jcm, 2022, doi:10.3390/jcm11123489_

Round 1

Reviewer 1 Report

This article by Zhou et al. demonstrated long term pregnancy outcome in women with diffuse adenomyosis after double-flap adenomyomectomy. This is a potentially interesting clinical study that is unaddressed properly and poorly described. I read this article with great interest and found that there are several critical issues that need to be resolved before further consideration.   

Major comments:

  1. In abstract, it is unclear whether successful pregnancy cases after surgery are due to natural conception or the result of ART.
  2. Introduction: the authors considered JZmax-A as an important factor to examine the pregnancy outcome. It is unclear the rationale why did they select this factor as a confounding factor. Is there any other study that indicate the low JZmax-A after surgery improves pregnancy rate. Please clarify this issue. 
  3. The measurement of JZmax-A is unclear and difficult to understand. Figure 1 is also not representative to measure JZmax-A. Please refer to the published articles on this issue (Kunz et al. 2005; Bazot et al. 2001). Please clarify in details in methods. 
  4. Increased women's age is a strong risk factor for fertility outcome. Fertility declines after 35 years of age and the chance of miscarriage increased. If this is true, then the authors should distribute the results of 56 pregnant women based on 35 or <35years versus >35years and all clinical profiles in these two groups of women and should represent them in a separate Table.  
  5. History of IRT failure is another factor in the fertility outcome in young women after surgery. this information is missing in Table 1. 
  6. The authors mentioned that about 42% of women with pregnant and non-pregnant women were coexistent with endometriosis. It is more important to examine the association between rASRM adhesion score (advanced endometriosis) and  fertility outcome in their pregnant women. This could be another weak point in this study. 
  7. The definition of clinical pregnancy rate is unclear in the methods. 
  8. In discussion, the authors indicated that their results are superior to other published reports. Are those reports were adjusted with  their own protocol. 
  9. In the groups of live birth, how many cases were delivered by Cesarian section and vaginal approach. This is unclear. 
  10. Difference in pregnancy outcome between natural conception and ART conception after surgery is unclear. 
  11. Discussion: Please make a short paragraph at the beginning with key findings and then discuss their relevance and irrelevance with other studies. 
  12. Discussion: The limitation section should come before final conclusion. The conclusion and key message to the readers based on their findings is weak and unclear. 
  13. The English language throughout the text is poor with many mistakes in grammar and the use of English syntax.

Minor comments: 

  1. Table 1. Please put n (%) in all incomplete data.
  2.  Table 1. Please mention median value and range of age and BMI
  3. Table 1. Please add history of ART failure  in each group. 

Reviewer 2 Report

Main comments

  1. With respect to surgery, double (or triple) flap surgery requires skilled surgeons and is difficult to implement in all types of facilities. Similarly, open laparoscopic surgery is often preferred, although laparoscopic-assisted adenomyomectomy can be offered, in some cases (please discuss).
  2. Given the fertility outcomes, alternative developing strategies (minimally invasive surgery, radiofrequency method, or new drug therapies for instance) should be given in the introduction and discussed accordingly.
  3. It is important to note that medical imaging is poorly introduced in the manuscript, even though it is an essential component of diagnosis and clinical management (see Bazot's review for example) and it plays central role in the results (JzMax-A).
  4. What was the recurrence rate after surgery, if any?
  5. Could the authors clarify whether surgery played an active role in stopping the desire to conceive in 24 patients?
  6. Please, clarify how the postoperative JZmax-A will be used in clinics as an indicator of pregnancy in case of unfavorable results: would the authors suggest another invasive surgery, suggest more ART, or deter any additional attempt?

Writing

Abstract: avoid abbreviations

Body of text: English language needs to be revised, including grammar and spelling (e.g., leisons, adenomayomectomy, we recruied). See also inconsistent punctuation.

Reviewer 3 Report

The manuscript is about pregnancy rate and outcome after double-flap adenomyomectomy in patients with diffuse adenomyosis. I found the manuscript interesting and very straight forward to understand, also pointing out important issues for gynecologists. I think it is almost ready for publication, only I will make few points for improvement.

What causes the difference in the thickness of JZ after operation. Discussion may be added about this. Is there an ideal operation method?

Line 218 “Thus, these results seemed to be very crucial for clinicians to provide recommendations to predict pregnancy after double-flap adenomyomectomy. “

I think there is no recommendation to make even if the pregnancy rate is low. If the authors mean with thick JZ, ART is preferable or have other recommendations, state more clearly.

Line 61 Misspelling of recruited

Too much data is repeated in the discussion, some numbers are not needed. 

I thought there may be some spaces missing. Please check again.

Reviewer 4 Report

Line 14 replace “junction” by “junctional” 

Line 54 to 62: The authors state that they used the technique of double flap for 10 years. However, they only recruited 143 patient. Did they study the different pregnancy rates between the early years and late years of practice? As with time, they are more experienced with the technique it might have an impact. 

Line 61 replace “recruited” by “recruited” 

Line 76 The authors used post op GnRha for 3 to 6 months. Do they have any concern about delayed healing with this medication? 

Line 82 the authors performed hysteroscopy for intra uterine adhesions, what was the incidence of these adhesions? 

Line 220 the strength of the study cannot be the conclusion itself. Strengths are reported as “prospective, double blinded, …”. The authors need to rephrase this paragraph. 

Line 230 as well it failed to prove the possible increase in pregnancy rate with the experience the surgeon will gain with time. 

Round 2

Reviewer 1 Report

I want to thank all authors of the article for their careful revision, properly responding to my previous comments and now the quality of this article is fairly improved. 

1. Figure 1. (A) dotted lines shows preoperative JZmax-A. Is the boundary of the dotted lines JZmax-A or size of the adeomyotic mass? because the line extends from the endometrium up to the outer myometrium. This is confusing and should clarified. I presume that JZ in the figure disappears and it is difficult to understanding thickness of JZ. (B) similarly reduced length of dotted lines indicates the reduced size of adenomyotic lesion and not the JZ thickness. This should be clarified.

2. Table 1 and Table 2. It is difficult to understand whether all data are represented in median, 95% CI, mean+/-SD or mean+/-SEM or number (%). Please add all these information against each parameter in each Table. Example, median, 95% CI for  age/BMI in Table 1 and mean+/-SD or no. (%) for other parameters in Table 1 and 2. In Table 2, % is missing for some parameters. Please correct them properly.

3. Please double check all grammatical and spelling mistakes throughout the the text.   

Reviewer 2 Report

The revised manuscript has been amended according to most of the referees' comments.

Some very interesting perspectives and elements of discussion (supported by appropriate references) are only shared in the rebuttal: I believe that this may also be presented in this manuscript for the clinical readership. I leave it to the discretion of the editor.

The form has been improved satisfactorily (few minor typos remain, see punctuation in Tables).

Author Response

Dear Reviewer ,

Thank you very much for your comments and suggestions. We have revised the manuscript again and responded point by point as follow.  Looking forward to hear from you soon.

 With kindest regards,

    Yours sincerely,

Xinmei Zhang

The revised manuscript has been amended according to most of the referees' comments.

Some very interesting perspectives and elements of discussion (supported by appropriate references) are only shared in the rebuttal: I believe that this may also be presented in this manuscript for the clinical readership. I leave it to the discretion of the editor.

The form has been improved satisfactorily (few minor typos remain, see punctuation in Tables).

Reply: We have made correction according to the Reviewer’s comments.